# Effects of Vaping Prevention Messages on Electronic Vapor Product Beliefs, Perceived Harms, and Behavioral Intentions among Young Adults: A Randomized Controlled Trial

**DOI:** 10.3390/ijerph192114182

**Published:** 2022-10-30

**Authors:** Andrea C. Villanti, Olivia A. Wackowski, S. Elisha LePine, Julia C. West, Elise M. Stevens, Jennifer B. Unger, Darren Mays

**Affiliations:** 1Rutgers Center for Tobacco Studies, New Brunswick, NJ 08901, USA; 2Department of Health Behavior, Society and Policy, Rutgers School of Public Health, Piscataway, NJ 08854, USA; 3Vermont Center on Behavior and Health, Department of Psychiatry, University of Vermont, Burlington, VT 05401, USA; 4Department of Psychology, University of Florida, Gainesville, FL 32611, USA; 5Department of Psychological Science, University of Vermont, Burlington, VT 05401, USA; 6Department of Population and Quantitative Health Sciences, Division of Preventive and Behavioral Medicine, University of Massachusetts Chan Medical School, Worcester, MA 01655, USA; 7Department of Preventive Medicine, Keck School of Medicine, University of Southern California, Los Angeles, CA 90032, USA; 8Department of Internal Medicine, The Ohio State University Wexner Medical Center, Columbus, OH 43210, USA; 9Center for Tobacco Research, The Ohio State University Comprehensive Cancer Center, Columbus, OH 43214, USA

**Keywords:** e-cigarette, vaping, randomized controlled trial, young adults, education, prevention

## Abstract

Youth have been the focus of electronic vapor product (EVP) prevention efforts though young adults had similar increases in current EVP use from 2015–2019. This study tested messages to reduce EVP use in young adults. Eight messages on vaping related harms and addictiveness combined with themes on social use and flavors were selected for inclusion in an online randomized controlled trial. Vermont young adults aged 18–24 (n = 569) were randomized to view the eight vaping prevention messages (n = 295) or eight messages on sun safety (n = 274). After completing baseline measures, participants viewed study messages and completed measures on message perceptions and perceived message effectiveness (PME), EVP-related beliefs, and EVP-related harm perceptions. Participants completed EVP-related beliefs and harm perception measures again at 1-month follow-up, as well as measures on tobacco and EVP-related behavioral intentions and behavior (ever and past 30-day use). Intervention participants reported positive impacts on vaping-related message responses. However, findings suggested no effect of vaping prevention messages on EVP-related beliefs, harm perceptions, or behaviors in the full sample. Exploratory analyses in the intervention condition showed that greater PME was associated with lower odds to intent to try cigarettes in the next year at follow-up.

## 1. Introduction

Successful mass media public education campaigns to prevent tobacco use have targeted knowledge and beliefs as the precursor to changing attitudes, and ultimately, tobacco use behavior [1,2,3,4,5]. Following the success of its Real Cost smoking prevention campaign [6,7,8], the U.S. Food and Drug Administration (FDA) developed a dedicated e-cigarette prevention messaging campaign largely deployed via social media [9,10]. Evaluations of FDA’s Real Cost e-cigarette campaign have documented high recall of prevention ads in youth, especially those who used social media at moderate or high frequencies [11] and associations between greater message exposure and higher odds of agreement with campaign-specific beliefs [12]. Other local, state, and national e-cigarette prevention campaigns, including those developed by Truth Initiative [13] and the American Lung Association [14], have not yet published evaluation data.

While youth have been the main focus of e-cigarette prevention efforts, young adults aged 18–24 have shown a high prevalence of electronic vapor product (EVP) experimentation since 2010 [15,16,17,18,19,20,21,22,23,24] and experienced increases in current exclusive e-cigarette use between 2015 and 2019 despite decreases in any tobacco use during this time [25]. Consistent with youth e-cigarette trends [26], young adults also had a 46% increase in current e-cigarette use prevalence between 2017 and 2018 [19]. However, the truth campaign is the only national effort that includes young adults in its target audience for vaping prevention efforts [27] and formative research for e-cigarette prevention messaging is largely focused on youth: to date, six studies have used randomized trials to test the effects of vape prevention messaging on message perceptions, risk perceptions, knowledge, and behavioral intentions and susceptibility [28,29,30,31,32,33], with only two of these targeted to young adults [29,31]. One of these studies examined the effect of anti-vaping public service announcements on young adult smokers and dual users [31], while the other exposed young adults to 16 gain- or loss- framed text messages on e-cigarette risks [29]. Together, these studies provide limited insight into how to address the main drivers of EVP use among young adults in educational messaging efforts.

In line with the Theory of Planned Behavior [34,35], we hypothesized that vaping prevention messages would alter beliefs, including harm perceptions, about EVPs, which would influence intention and use of EVPs. This is consistent with previous literature showing that young adults’ EVP use is influenced by perceptions that e-cigarettes are less harmful or addictive than cigarettes or other products [36,37,38], are socially acceptable [27,36,39,40], are easy to use [23], and come in appealing flavors [39,41]. However, content themes included in the existing six vape prevention randomized trials were limited to health-related harms, addiction, and anti-industry messages [28,29,30,31,32,33]. The current study used a multi-phase design to develop and test a series of vaping prevention messages that could be delivered via social media (e.g., Instagram) to address key factors associated with e-cigarette use in young adults: (1) Harm Perceptions, (2) Addictiveness, (3) Social Use, and (4) Flavors. We first developed a series of harm and addiction messages which were combined with themes related to social aspects of use and flavors and tested in two randomized experiments (Message Optimization phase) published previously [42]. Selected messages from this optimization phase were then used as the vape messaging prevention intervention in the current study, an online randomized controlled trial which compared these messages with a similar number of control messages on sun safety. This manuscript reports on outcomes from the randomized controlled trial. Consistent with our prior work in testing other types of tobacco messaging interventions [43,44,45], we hypothesized that exposure to the vape messaging condition would result in greater agreement with nicotine vaping-related beliefs and higher perceived harm of EVPs. We also explored the durability of effects on vaping-related beliefs and behavior at 1-month follow-up.

## 2. Materials and Methods

### 2.1. Trial Design

The PACE Vape Messaging study was a parallel, two-group individually randomized controlled trial conducted in one state in the United States comparing exposure to vape education messages (intervention) to exposure to messages on sun safety (control), with all messages formatted for social media. Enrolled participants completed a baseline survey in August 2020 in which they viewed eight intervention or control messages, then completed outcome measures immediately post-exposure and again at one-month follow-up in September 2020. We based our sample size calculations on a conservative scenario for the intervention effects: a 10.4% relative increase in agreement with campaign-targeted beliefs about vaping-related risks in the intervention compared to the control group, consistent with effects seen in the evaluation of FDA’s Real Cost media campaign [6]. With 604 participants (n = 302 in each group), the study was powered to detect a 10.4% difference in campaign-targeted beliefs between groups with 80% power and two-tailed alpha = 0.05. The study was approved by the University of Vermont Institutional Review Board and registered on ClinicalTrials.gov (NCT04450537).

### 2.2. Participants

Participants were recruited from June through August 2020 using online advertisements (e.g., Facebook, Google, Snapchat), existing social media platforms for the Policy and Communication Evaluation (PACE) Vermont study (pacevt.org, @pace_vt), a statewide community e-mail digest in Vermont (Front Porch Forum), partner referrals, and past participants in the PACE Vermont Cohort Study, a multi-wave survey study of substance use beliefs and behaviors in Vermont youth and young adults [46]. Eligible participants were Vermont residents aged 18–24 who had access to a smartphone with internet access and used one or more social media site(s) at least weekly. This study was advertised using branding from the broader PACE Vermont Cohort Study, and participants were simultaneously recruited for this messaging study and for additional waves of the PACE Vermont Cohort Study, funded by a separate grant from the National Institute on Drug Abuse (R21DA051943). Online ads directed participants to a study website and a brief online Qualtrics survey assessing their interest in the study, their age group, their e-mail and cell phone number and preferred mode of contact; automated processes in Qualtrics screened out bots or likely fraudulent responses. Upon valid completion of the brief interest survey, participants immediately received a unique link to the screener survey to assess eligibility by e-mail or text message, depending on their preference. The addition of this step ensured valid contact information for participants. Potentially eligible participants were then asked to provide consent to participate in the study and complete a short consent quiz to ensure understanding of study procedures. Participants who were eligible and consented to the study were directed to complete an online payment form in compliance with University financial policies. After screening and consent, participants’ responses were screened manually by study staff to filter out potentially deceptive responses, including: (1) conducting consistency checks between age and date of birth, as well as state of residence and location of IP address; (2) adding a CAPTCHA item in the screener to ensure that respondents were human and not bots; (3) conducting additional screening of respondents with suspicious email addresses (e.g., common e-mail format across surveys completed within minutes of each other and email addresses including names that did not correspond to contact information) and out-of-state phone numbers; and (4) using information from the screening and payment forms (e.g., consistency of name across forms and location of participant address) [47]. Suspicious participants were notified of their ineligibility with the opportunity to refute the decision by providing additional information. Eligible young adults who consented to participate in both the PACE Vape Messaging Study and Waves 4–6 of the PACE Vermont Cohort Study were able to do so and skip logic was employed in the Wave 4 survey instrument to minimize duplication of items across the two studies.

### 2.3. Study Conditions

Participants in the Messaging Trial were randomized via Qualtrics at baseline in a 1:1 allocation to either the intervention or control condition in which they viewed eight messages; the order of the messages remained the same in each condition. Each message was programmed to have a minimum viewing time of five seconds with an embedded heatmapping task where participants were asked to identify up to five areas of the message that attracted their attention by pointing and clicking on those areas of the image. The resulting exposure to each condition lasted at least 40 seconds across the eight messages.

#### 2.3.1. Intervention Condition

The vape education messages used in this trial were developed and selected from a two-phase optimization study that first assessed the likeability and perceived message effectiveness of 32 images and 33 messages using a 2 (content: addiction, harm) × 3 (theme: alone, + flavors, + social) design, then paired the 24 most effective messages with 6 images rated most likeable and 6 images rated most effective at discouraging vaping [42]. Based on results of this previous formative study, we selected the eight vaping education messages (overlaid on images in a format similar to that used on social media channels such as Instagram) that produced the highest perceived message effectiveness for use in this trial: two messages on harms alone, two on harms + social, two on harms + flavor, one on addiction + social, and one on addiction + flavor (Appendix A).

#### 2.3.2. Control Condition

Control participants were exposed to eight messages on sun safety used in a previous trial [43], but reformatted for social media to ensure equal attention to study messages, without an anticipated impact on the outcomes of interest (Appendix A).

### 2.4. Measures

#### 2.4.1. Baseline Characteristics

All participants were asked to provide information on age, sex assigned at birth, sexual identity, race, Hispanic ethnicity, highest level of education completed, and subjective financial status, a validated measure of socioeconomic status for young adults [48]. These characteristics were chosen due to their correlations with tobacco use in young adults [49,50,51]. Participants were also asked about awareness of national tobacco prevention media campaigns (i.e., FDA’s Real Cost, truth) and a local vaping prevention digital media campaign (Unhyped). Frequency of exposure to tobacco advertisements on the internet, in newspapers or magazines, or on TV, streaming services, or the movies was assessed using a six-point Likert scale with the following options: (1) I do not [use the internet/read newspapers or magazines; watch TV or streaming services, or go to the movies]; (2) Never; (3) Rarely; (4) Sometimes; (5) Most of the time; or (6) Always.

#### 2.4.2. Tobacco Use

Ever use of cigarettes and electronic vapor products (EVPs) was assessed among all participants at baseline and follow-up, as well as past 30-day use of cigarettes, EVPs, smokeless tobacco (chewing tobacco, snuff, dip, snus, or dissolvable tobacco products), cigars (cigars, cigarillos, or little cigars), and hookah (hookah or waterpipe). Introductory wording in the EVP section of the survey highlighted that these questions related to using nicotine liquids, pods, or cartridges in electronic vapor products. Changes in ever use between baseline and follow-up captured trial among non-users during the study period. We also assessed past-year and past-month attempts to quit or cut down on cigarette and EVP use at baseline; at follow-up, we assessed only past-month attempts to quit or cut down on these products.

#### 2.4.3. Response to Study Messages

Following exposure to all eight intervention or control messages, several items were asked to assess message receptivity and potential impact. These items addressed message relevance and who the messages were directed to (someone like you, other types of people, or a mix), as well as the value of information in the messages. “Likeability” of the message, which has strong predictive power for advertising success [52,53], was assessed by asking participants to describe their feelings about the messages, with responses on a 5-point scale ranging from “I liked them very much” to “I disliked them very much.” Two items also assessed overall message ratings on a scale from “poor” (1) to “excellent” (6) and whether the messages provided participants with new information. Vaping-related responses to the messages were collected in two ways: as cognitive reactions and as potential impact on behavior. Cognitive reactions were assessed as perceived message effectiveness (PME), using a validated three-item scale of effects perceptions [54]. These items were: “These messages discourage me from wanting to vape” (discouragement), “These messages make me concerned about the health effects of vaping” (concern), and “These messages make vaping seem unpleasant to me” (unpleasantness). Response options were on a five-point Likert scale, from strongly disagree (1) to strongly agree (5) and the mean response was calculated per participant for each message. [54]. Two additional items addressed potential behavioral responses to the messages as their effect on curiosity to try vape products and desire to quit or cut down on vape products. Response options to these two items were “increase,” “decrease,” or “no effect.” Dwell time on the messages was collected passively for each message by Qualtrics and summed to estimate total duration of exposure to the messages in each study condition.

#### 2.4.4. Outcomes

Our primary outcomes were nicotine-related EVP beliefs and harm perceptions, post-exposure and at follow-up. At baseline, these items were assessed after participants viewed all eight messages in their assigned study condition. Nicotine-related EVP beliefs were “Nicotine is the main substance in electronic vapor products that makes people want to vape”, “One 5% vape pod can contain as much nicotine as an entire pack of cigarettes”, and “The claim that a vape is low in nicotine means that it is less addictive”, with three response options (true, false, don’t know) in line with our previous work [47,55,56]. We adapted items from the Population Assessment of Tobacco and Health (PATH) Study to assess absolute harm perceptions of vaping and the relative harm of vaping compared to regular cigarettes in line with our work [57]; we added an item on relative harm of vaping nicotine compared with vaping marijuana/THC, given the e-cigarette and vaping-related lung injury (EVALI) epidemic in 2019. The absolute harm item was “How much do you think people harm themselves when using electronic vapor products?” with response options (“No harm,” “A little harm,” “Some harm,” and “A lot of harm”) and two relative harm items (“Is using electronic vapor products less harmful, about the same or more harmful than smoking cigarettes?” and “Is vaping nicotine less harmful, about the same or more harmful than vaping marijuana/THC?”) with options “Less harmful,” “About the same,” and “More harmful.” Given message content on flavors, we also asked about relative harm of flavored versus unflavored tobacco products and e-cigarettes, with response options “Less harmful,” “No different,” and “More harmful.”

Our secondary outcomes were EVP-related norms, vaping-related behavioral intentions and use behaviors at follow-up. Two items on EVP-related norms were adapted from the PhenX Toolkit (#750301) on Social Norms about Tobacco: “How would you describe most people’s opinion of using electronic vapor products like JUUL?” and “Thinking about the people who are important to you, how would you describe their opinion of using electronic vapor products like JUUL?” with response options “Very positive,” “Somewhat positive,” Neither positive nor negative,” “Somewhat negative,” and “Very negative.” Four items, adapted from the PhenX Toolkit (#710302) assessed behavioral intention to use an EVP in the next 12 months. EVP use behaviors included trial (among non-users at baseline), past 30-day use, and among ever EVP users, attempts to quit or cut down on vaping in the past month. We also collected these measures for cigarette use for comparison.

Other beliefs assessed at baseline and follow-up were drawn from our previous work [43,56,58,59] and were included to assess potential spillover or unintended effects of the messaging, including “Nicotine is a cause of cancer,” “A tobacco product that says it has no additives is less harmful than a regular tobacco product,” “A tobacco product that says it is organic is less harmful than a regular tobacco product” with response options “True,” “False,” and “Don’t know.”

### 2.5. Statistical Analysis

Analyses conducted in 2022 used Stata MP, Version 17.0, and examined distributional properties of all variables and used t-tests and/or chi-square tests to identify any differences between study conditions on demographic or vaping-related characteristics at baseline or in those lost-to-follow-up. This ensured baseline equivalence, and when differences were found at the *p* < 0.10 level, outcome models included those covariates. We examined message response (including differences in message dwell time, likeability, relevance, novelty and vaping related responses) by study condition using t-tests and chi-square tests. We used similar methods to examine differences in vaping-related nicotine beliefs, harm perceptions, norms, and our manipulation checks immediately post-exposure and at 1-month follow-up using per protocol analyses. Multivariable logistic regression models were developed to estimate the odds of behavioral intention and behavioral outcomes by study condition at 1-month follow-up, controlling for baseline measures of the outcome and baseline exposure to the Real Cost campaign, on which there was imbalance in the sample at follow-up. Exploratory subgroup analyses examined the role of message engagement on study outcomes, assessing relationships between dwell time, PME, and vaping-related beliefs, harm perceptions, norms, and manipulation checks in the intervention condition only. These analyses used logistic and linear regression models in which dwell time or PME was the exposure of interest and vaping-related measures were the outcomes of interest; multinomial logistic regression models were used for relative harm perception outcomes with three levels (i.e., “less harmful”, “no different”, “more harmful) with “no different” serving as the reference category. Based on the multiple outcomes assessed, we used a Bonferroni correction to adjust our threshold for statistical significance (α set at 0.05/47 models = Bonferroni corrected α 0.001) for all multivariable models presented.

## 3. Results

From June to August 2020, 2680 participants were screened and in August 2020, 569 were randomized to either the vape education intervention (n = 295) or control (n = 274) condition (Figure 1), with 294 participants receiving the allocated intervention and 272 receiving the allocated control. Overall follow-up at 1-month (September 2020) was 90% (91% intervention vs. 90% control, *p* = 0.38). Those retained in the study were generally similar to those lost to follow-up, though a greater proportion of those lost to follow-up were female, identified as white, had ever used a cigarette at baseline, and reported past 30-day smokeless tobacco use at baseline compared to those retained (Appendix A). Generally, there remained balance in these characteristics by study condition in the retained sample with the only differences at the *p* < 0.10 level seen in past 30-day smokeless tobacco use and awareness of the FDA’s Real Cost tobacco prevention campaign (Appendix A). Multivariable models of measures at 1-month follow-up controlled for baseline exposure to the Real Cost campaign to adjust for this imbalance, though given the small number of participants reporting smokeless tobacco use (n = 5), we did not include past 30-day smokeless tobacco use as a covariate in those models.

The mean age of participants was 21.1 years and most were female (70%), identified as cisgender (94%), straight or heterosexual (69%), and white (76%; Table 1). Nearly half of participants lived in a rural county (48%), 36% worked full-time, and most reported subjective financial status of living comfortably (34%) or meeting needs with a little left (38%). More than half had ever used an EVP (63%), compared with 37% who had ever used cigarettes. In the full sample, 29% reported past 30-day EVP use and 16% past 30-day cigarette use. Exposure to national tobacco prevention campaigns was 66% for FDA’s Real Cost and 56% for truth, with only 21% reporting awareness of Vermont’s vaping prevention campaign (Unhyped).

### 3.1. Effect of Study Condition on Message Responses

Participants in the intervention condition spent nearly 2.7 min of dwell time on study messages compared with 2.2 min in the control condition (Table 2). Message relevance was higher for the control (sun safety) messages than for intervention messages and a greater proportion of intervention participants felt that study messages were directed to others (38% vs. 21%; *p* < 0.001) and not valuable (25% vs. 14%; *p* = 0.001) compared with control participants. Likeability, overall message rating, and novelty of study messages were similar across conditions.

Vaping-related responses to messages was higher in the intervention than control condition, including higher mean PME ratings (3.7 vs. 2.5) and greater perceived impact on decreasing curiosity to vape (56% vs. 28%) and increasing desire to quit or cut down on vaping (34% vs. 15%; Table 2).

### 3.2. Effect of Study Condition on Primary Outcomes: Vaping-Related Beliefs and Harm Perceptions

There was generally high agreement that “nicotine is the main substance in EVPs that makes people want to vape” (88%), that “one 5% vape pod contains as much nicotine as a pack of cigarettes” (75%), and that “addiction to nicotine is something I am concerned about” (74%) immediately post-exposure; these beliefs remained prevalent at follow-up (Table 3). Similarly, more than 70% of the sample endorsed that EVPs carried “some” or “a lot” of harm and that EVPs were as or more harmful than smoking cigarettes both post-exposure and at follow-up. More than 40% believed that vaping nicotine was more harmful than vaping marijuana. There were no effects of intervention exposure on nicotine-related vaping beliefs or harm perceptions immediately post-exposure or at one-month.

### 3.3. Effect of Study Condition on Secondary Outcomes: Vaping-Related Norms, Behavioral Intentions and Behaviors

There were no effects of intervention condition on EVP or cigarette-related behavioral intentions or behaviors at 1-month follow-up in logistic regression models that controlled for baseline intentions or behaviors, depending on the outcome modeled (Table 4).

### 3.4. Effect of Study Condition on Potential Spillover Effects

While there were no effects of intervention exposure on primary or secondary outcomes immediately post-exposure or at one-month follow-up, participants in the intervention condition were more likely to endorse nicotine as a cause of cancer (69%) compared with control participants (59%, *p* = 0.03) immediately post-exposure; there was no difference between conditions at follow-up. Similarly, intervention participants did not differ from control participants in their immediate post-exposure responses to “a tobacco product that says it has no additives is less harmful than a regular tobacco product”, but at 1-month follow-up, intervention participants were more likely to report this as true (17% vs. 12%) or “don’t know” (26% vs. 19%; *p* = 0.03).

### 3.5. Exploratory Analyses of the Role of Message Engagement on Study Outcomes

Exploratory subgroup analyses in the intervention participants examined whether message engagement (i.e., dwell time, PME) affected vaping-related beliefs, harm perceptions, norms, behavioral intentions, and behavior immediately post-exposure and at 1-month follow-up (Table 5). Dwell time was not associated with any of the outcome measures. There were few associations between PME and study outcomes, though they suggested that a one-unit increase in PME was positively associated with greater concern about addiction to nicotine (OR 1.44, 95% CI 1.10–1.90) and lower likelihood of endorsing EVPs as “less harmful” than smoking cigarettes (vs. “no different”; RRR 0.64, 95% CI 0.48–0.85) and vaping nicotine as “less harmful” than vaping marijuana/THC (RRR 0.63, 95% 0.45–0.87) immediately post-exposure; however, these findings were not significant after adjusting for multiple comparisons. At 1-month follow-up, a one-unit increase in PME was associated with higher perceived harm of EVPs (b 0.14, 95% CI 0.051–0.22) and lower odds of a correct response to nicotine is a cause of cancer (OR 0.62, 95% CI 0.45–0.86); again, these were not significant after adjusting for multiple comparisons.

There were no associations between PME and behavioral intentions related to EVP use at follow-up in the subset of intervention participants who had not tried an EVP at baseline, controlling for baseline intentions. A one unit increase in PME was associated with lower odds at 1-month follow-up of trying a cigarette in the next year (OR 0.47, 95% CI 0.31–0.72) in intervention participants who had not tried cigarettes at baseline, controlling for baseline measures of each outcome. Similarly, there were no associations between PME and past 30-day EVP, cigarette, or cigar use at 1-month follow-up in intervention participants.

## 4. Discussion

Despite the prevalence of e-cigarette use in young adults [19,25], few vaping prevention efforts have been developed and tested for this age group. This study used a randomized design to evaluate the immediate and short-term effects of a vape messaging intervention on vaping-related beliefs, harm perceptions, norms, behavioral intentions, and behaviors in young adults. Findings suggested that the intervention produced the desired response on message-related outcomes of perceived message effectiveness, reduced curiosity about vaping, and increased desire to quit or cut down on EVPs. However, there was no impact of the intervention on EVP-related beliefs or harm perceptions. Analyses exploring the potential associations between perceived message effectiveness and EVP-related outcomes in the intervention group supported that higher PME was associated with higher odds of being concerned about addiction to nicotine immediately post-exposure and with higher perceived harm of EVPs at follow-up, consistent with research conducted in adolescents [33]; however, these findings were not significant after adjusting for multiple comparisons. Higher PME was not associated with EVP-related behavioral intentions, but it was associated with lower intention to try cigarettes in the next year; this could be related to proportion of never cigarette users (62%) versus never EVP users (38%) in the intervention condition. Findings suggested that higher PME may be prospectively associated with lower odds of believing that nicotine is not a cause of cancer. While our messages did not address the relationship between nicotine and cancer or additive-free EVPs, they did include statements that “vapes contain ingredients that can cause cancer” and “chemicals in vapes can cause cell and lung damage.” Both of these messages may have inadvertently created mental maps between nicotine and cancer and between vape additives and health harms. These findings highlight that prevention messages addressing the harms of nicotine and other constituents in EVPs should be rigorously tested to ensure that they do not contribute to false beliefs about nicotine, EVPs, and other tobacco products. They also suggest that identifying effective messages as those scoring above the midpoint on message perceptions scales [1,60,61] may not be appropriate as the primary measure of effectiveness where greater nuance in tobacco messaging is needed.

Current randomized trials of vape messaging have addressed some, but not all, of the outcomes assessed in our trial [28,29,30,31,32,33]. Only two vape prevention message trials included measures of PME [30,33] with one showing desired relationships between PME, risk beliefs about vaping, attitudes toward vaping, and intentions to vape [33]. Three of the studies assessed behavioral intentions [31,32,33], with two showing that message exposure reduced intentions to vape or purchase EVPs [31,32]. No formative studies have tested the effects of vape prevention messaging on e-cigarette use behaviors. The comprehensiveness of our measures provides a more complete picture of message effects on a range of outcomes and the potential that vaping prevention interventions producing desired responses to the messages may have no impact on EVP-related harm perceptions or behaviors. However, our use of message response items immediately after intervention exposure likely contaminated our assessment of target outcome measures; future studies may consider conducting outcome assessment immediately following exposure or decoupling message response items from outcome measures in the same assessment. High awareness of other national tobacco prevention efforts in our sample and the timing of the trial in the year following the e-cigarette and vaping-associated lung injury (EVALI) epidemic and during the COVID pandemic may have reduced any potential differences in response between study conditions, as participants may have changed their vaping beliefs, harm perceptions, and behaviors surrounding these secular events [62,63,64,65,66,67]. As a result, our participants may have had little room to change on our targeted beliefs and thus, we found no effect of our vaping prevention messages on beliefs

Strengths of the current study include the focus on young adults, randomized study design with an attention control condition, high retention, and use of an evidence-informed series of messages in the intervention condition [42]. The study is limited by the nature of the online convenience sample conducted in a single state, the high proportion of females in the study, and the lack of data collected on outcomes directly related to intervention messages, including perceived risk or harm of lung damage and impact of vaping on mental health. The higher proportion of females enrolled in this is consistent with other web-based health studies using online advertising [68,69,70,71], including our earlier work in Vermont young adults [45]. Data from the National Health Interview Study report higher prevalence of EVP use among male young adults [19] and our findings may have been affected by the imbalance by sex. Notably, however, the prevalence of past 30-day e-cigarette use in our sample is approximately five times higher than reported in the NHIS and more consistent with results for young adults in the Population Assessment of Tobacco and Health [72]; thus, effects of our intervention on behavior are likely to be generalizable to the broader population. Results from this study, limited to a single point in time message exposure, included a range of messages that could be used in a media campaign to sustain awareness and drive behavior change, but the assessment of message responses for the intervention as a whole and not for individual messages limits our ability to refine the current intervention for greater impact. Another consideration is that the control condition addressed a health-related topic and attention to these sun safety messages may have stimulated participants in the control condition to think about health risks and impacted response to vaping-related outcome measures, though this was not the case in our prior study of nicotine messaging in a sample of adults [43].

## 5. Conclusions

A vape messaging intervention designed for U.S. young adults demonstrated positive impacts on vaping-related message responses, including higher perceived message effectiveness, decreased curiosity to vape, and increased desire to quit or cut down on vaping compared with the control condition. The intervention did not, however, produce greater impacts on vaping-related beliefs, harm perceptions, norms, behavioral intentions, or behaviors than the sun safety control messages immediately post-exposure or at 1-month follow-up. These findings are consistent with the limited effects of a national tobacco education campaign targeting young adults that used a randomized design and digital delivery of messages [73]. In line with prior research on effective message themes for EVP prevention efforts [27,74,75], the intervention largely addressed health risks associated with constituents in EVPs [42], though it also included two messages related to addiction, which has been shown to be a less effective messaging theme in young people. Findings from this study suggest that vaping messages addressing the harms of nicotine and other EVP ingredients may increase false beliefs about the health harms of nicotine and additive-free tobacco products. Future studies are needed to better understand the role of message perceptions and message response items in testing novel messages for tobacco prevention efforts, as well as how to design effective prevention messages about EVPs without producing unintended impacts on false beliefs about nicotine and other tobacco products.

## Figures and Tables

**Figure 1 ijerph-19-14182-f001:**
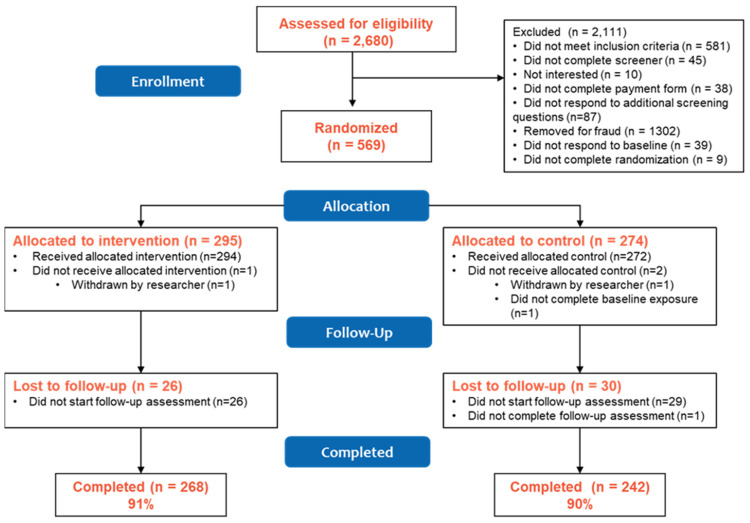
PACE Vape Messaging Study CONSORT Diagram.

**Table 1 ijerph-19-14182-t001:** Sample characteristics, PACE Vape Messaging Study, 2020.

	Control n (%)	Intervention n (%)	Total n (%)	*p*-Value
Age (mean (SD))	21.1 (2.0)	21.2 (1.9)	21.1 (1.9)	0.76
Sex				0.76
Male	80 (29.5)	90 (30.7)	170 (30.1)	
Female	191 (70.5)	203 (69.3)	394 (69.9)	
Gender identity				0.24
Cisgender	259 (95.2)	273 (92.9)	532 (94.0)	
Transgender/don’t know/questioning	13 (4.8)	21 (7.1)	34 (6.0)	
Sexual identity, not heterosexual vs. heterosexual				0.70
Straight/heterosexual	191 (70.2)	202 (68.7)	393 (69.4)	
Not straight/heterosexual	81 (29.8)	92 (31.3)	173 (30.6)	
Race/ethnicity, 3 categories				
White	208 (76.5)	224 (76.2)	432 (76.3)	0.99
Non-white/other race	24 (8.8)	27 (9.2)	51 (9.0)	
Hispanic	40 (14.7)	43 (14.6)	83 (14.7)	
HRSA-designated rural county				0.99
No	137 (51.7)	146 (51.8)	283 (51.7)	
Yes	128 (48.3)	136 (48.2)	264 (48.3)	
Employment status				0.16
Work full-time (35 h/week or more)	100 (36.8)	103 (35.0)	203 (35.9)	
Work part-time (15–34 h/week)	70 (25.7)	57 (19.4)	127 (22.4)	
Work part-time (<15 h/week)	39 (14.3)	56 (19.0)	95 (16.8)	
Don’t currently work for pay	63 (23.2)	78 (26.5)	141 (24.9)	
Subjective financial status, YA only				0.38
Live comfortably	95 (34.9)	99 (33.7)	194 (34.3)	
Meet needs with a little left	99 (36.4)	118 (40.1)	217 (38.3)	
Just meet basic expenses	74 (27.2)	68 (23.1)	142 (25.1)	
Don’t meet basic expenses	4 (1.5)	9 (3.1)	13 (2.3)	
Ever use				
Cigarettes	96 (35.3)	114 (38.8)	210 (37.1)	0.39
Electronic vapor products (EVP)	173 (63.6)	185 (62.9)	358 (63.3)	0.87
Past 30-day use				
Cigarettes	37 (13.7)	53 (18.2)	90 (16.0)	0.15
Electronic vapor products (EVP)	75 (27.6)	90 (30.6)	165 (29.2)	0.43
Cigar/cigarillo/little cigar	17 (6.3)	16 (5.4)	33 (5.8)	0.68
Smokeless tobacco	5 (1.8)	3 (1.0)	8 (1.4)	0.41
Hookah or waterpipe	7 (2.6)	5 (1.7)	12 (2.1)	0.47
Exposure to tobacco prevention campaigns				
The Real Cost (FDA)	169 (62.1)	206 (70.1)	375 (66.3)	0.14
truth	147 (54.0)	169 (57.5)	316 (55.8)	0.28
Unhyped	50 (18.4)	68 (23.1)	118 (20.8)	0.05
Frequency of seeing ads or promotions for cigarettes or other tobacco products				
On the internet (mean (SD))	2.15 (0.78)	2.13 (0.82)	2.14 (0.80)	0.79
In newspapers or magazines (mean (SD))	1.71 (1.07)	1.65 (1.12)	1.68 (1.09)	0.54
On TV or streaming services (mean (SD))	1.76 (0.75)	1.77 (0.76)	1.77 (0.75)	0.99

**Table 2 ijerph-19-14182-t002:** Message response, by study condition, PACE Vape Messaging Study, 2020.

	Control n (%)	Intervention n (%)	Total n (%)	*p*-Value
Dwell time on messages in seconds (mean (SD))	133.13 (138.79)	162.89 (164.05)	148.59 (153.02)	0.02
Message perceptions, 1–5 scale (mean (SD))				
Relevance	3.19 (1.12)	2.58 (1.26)	2.88 (1.23)	<0.001
Likeability	3.28 (0.93)	3.22 (0.96)	3.25 (0.94)	0.52
Overall message rating, 1–6 scale (mean (SD))	3.97 (1.10)	3.95 (1.27)	3.96 (1.19)	0.83
Direction of messages				<0.001
Directed to me	70 (25.7)	94 (32.0)	164 (29.0)	
Directed to others	58 (21.3)	114 (38.8)	172 (30.4)	
Mix	144 (52.9)	86 (29.3)	230 (40.6)	
Value of messages				0.001
Mostly valuable	82 (30.1)	96 (32.8)	178 (31.5)	
Not valuable	38 (14.0)	74 (25.3)	112 (19.8)	
Mix	152 (55.9)	123 (42.0)	275 (48.7)	
Message provide new information				0.071
No	124 (45.6)	112 (38.1)	236 (41.7)	
Yes	148 (54.4)	182 (61.9)	330 (58.3)	
**Vaping-related responses**				
Perceived message effectiveness, 1–5 scale (mean (SD))	2.54 (1.32)	3.72 (0.97)	3.15 (1.29)	<0.001
Messages’ effect on curiosity to vape				<0.001
No effect	192 (70.6)	120 (40.8)	312 (55.1)	
Increase	4 (1.5)	9 (3.1)	13 (2.3)	
Decrease	76 (27.9)	165 (56.1)	241 (42.6)	
Messages’ effect on desire to quit/cut down vaping				<0.001
No effect	204 (75.0)	162 (55.1)	366 (64.7)	
Increase	40 (14.7)	100 (34.0)	140 (24.7)	
Decrease	28 (10.3)	32 (10.9)	60 (10.6)	

**Table 3 ijerph-19-14182-t003:** Effect of study condition on vaping-related beliefs, harm perceptions and norms immediately post-exposure and at 1-month follow-up, PACE Vape Messaging Study, 2020.

	Post-Exposure				1-Month Follow-Up			
	Control n (%)	Intervention n (%)	Total n (%)	*p*-Value	Control n (%)	Intervention n (%)	Total n (%)	*p*-Value
**Vaping-related beliefs**								
Nicotine is main substance in EVPs that makes people want to vape				0.11				0.80
False	11 (4.1)	14 (4.8)	25 (4.4)		13 (5.4)	11 (4.1)	24 (4.7)	
True	245 (90.4)	250 (85.0)	495 (87.6)		204 (84.6)	229 (85.8)	433 (85.2)	
Don’t know	15 (5.5)	30 (10.2)	45 (8.0)		24 (10)	27 (10.1)	51 (10)	
One 5% vape pod contains as much nicotine as pack of cigarettes				0.22				0.95
False	13 (4.8)	15 (5.1)	28 (5.0)		7 (2.9)	7 (2.6)	14 (2.7)	
True	196 (72.3)	229 (77.9)	425 (75.2)		193 (79.8)	212 (79.1)	405 (79.4)	
Don’t know	62 (22.9)	50 (17.0)	112 (19.8)		42 (17.4)	49 (18.3)	91 (17.8)	
Addiction to nicotine is something I am concerned about				0.39				0.88
False	57 (21.0)	53 (18.0)	110 (19.4)		62 (25.6)	71 (26.5)	133 (26.1)	
True	194 (71.3)	224 (76.2)	418 (73.9)		162 (66.9)	180 (67.2)	342 (67.1)	
Don’t know	21 (7.7)	17 (5.8)	38 (6.7)		18 (7.4)	17 (6.3)	35 (6.9)	
**Vaping-related harm perceptions**								
Harm from EVPs				0.81				0.31
No harm	1 (0.4)	3 (1.0)	4 (0.7)		1 (0.4)	1 (0.4)	2 (0.4)	
A little harm	38 (14.0)	43 (14.6)	81 (14.3)		22 (9.1)	35 (13.1)	57 (11.2)	
Some harm	150 (55.1)	157 (53.4)	307 (54.2)		131 (54.1)	152 (56.7)	283 (55.5)	
A lot of harm	83 (30.5)	91 (31.0)	174 (30.7)		88 (36.4)	80 (29.9)	168 (32.9)	
Harm of EVPs vs. smoking cigarettes				0.50				0.72
Less harmful	71 (26.1)	88 (29.9)	159 (28.1)		48 (19.8)	49 (18.3)	97 (19)	
No different	143 (52.6)	141 (48.0)	284 (50.2)		146 (60.3)	171 (63.8)	317 (62.2)	
More harmful	58 (21.3)	65 (22.1)	123 (21.7)		48 (19.8)	48 (17.9)	96 (18.8)	
Harm of vaping nicotine vs. marijuana				0.56				0.97
Less harmful	61 (22.4)	58 (19.7)	119 (21.0)		46 (19)	49 (18.3)	95 (18.6)	
No different	103 (37.9)	107 (36.4)	210 (37.1)		88 (36.4)	100 (37.3)	188 (36.9)	
More harmful	108 (39.7)	129 (43.9)	237 (41.9)		108 (44.6)	119 (44.4)	227 (44.5)	
**Vaping-related norms (1 (very positive) to 5 (very negative); mean (SD))**								
Most people’s opinion of using EVPs like JUUL	2.81 (0.99)	2.85 (1.07)	2.83 (1.03)	0.61	2.87 (0.99)	2.95 (1.03)	2.91 (1.01)	0.34
Opinion of people who are important to you of using EVPs like JUUL	3.48 (1.08)	3.40 (1.11)	3.44 (1.09)	0.38	3.42 (1.13)	3.46 (1.07)	3.44 (1.10)	0.70
**Potential spillover effects**								
Nicotine is cause of cancer				0.03				0.50
False	62 (22.8)	55 (18.7)	117 (20.7)		53 (22.0)	52 (19.4)	105 (20.6)	
True	161 (59.2)	204 (69.4)	365 (64.5)		149 (61.8)	179 (66.8)	328 (64.4)	
Don’t know	49 (18.0)	35 (11.9)	84 (14.8)		39 (16.2)	37 (13.8)	76 (14.9)	
Ease of use of flavored tobacco/EVPs vs. unflavored				0.19				0.17
Easier to use	201 (73.9)	207 (70.6)	408 (72.2)		176 (72.7)	183 (68.3)	359 (70.4)	
About the same	64 (23.5)	83 (28.3)	147 (26.0)		66 (27.3)	82 (30.6)	148 (29)	
Harder to use	7 (2.6)	3 (1.0)	10 (1.8)		0 (0)	3 (1.1)	3 (0.6)	
Harm of flavored tobacco/EVPs vs. unflavored				0.20				0.25
Less harmful	7 (2.6)	12 (4.1)	19 (3.4)		11 (4.5)	6 (2.2)	17 (3.3)	
No different	209 (76.8)	207 (70.4)	416 (73.5)		181 (74.8)	197 (73.5)	378 (74.1)	
More harmful	56 (20.6)	75 (25.5)	131 (23.1)		50 (20.7)	65 (24.3)	115 (22.5)	
A tobacco product that says it has no additives is less harmful than a regular tobacco product				0.38				0.03
False	140 (51.5)	166 (56.5)	306 (54.1)		166 (68.6)	153 (57.1)	319 (62.5)	
True	49 (18.0)	53 (18.0)	102 (18.0)		29 (12)	45 (16.8)	74 (14.5)	
Don’t know	83 (30.5)	75 (25.5)	158 (27.9)		47 (19.4)	70 (26.1)	117 (22.9)	
A tobacco product that says it is organic tobacco is less harmful than a regular tobacco product				0.48				0.51
False	169 (62.1)	196 (66.7)	365 (64.5)		179 (74)	186 (69.4)	365 (71.6)	
True	39 (14.3)	34 (11.6)	73 (12.9)		22 (9.1)	27 (10.1)	49 (9.6)	
Don’t know	64 (23.5)	64 (21.8)	128 (22.6)		41 (16.9)	55 (10.5)	96 (18.8)	

**Table 4 ijerph-19-14182-t004:** Effect of study condition on behavioral intentions and behavior at 1-month follow-up, PACE Vape Messaging Study, 2020.

Outcomes	n	Odds Ratio (95% CI) ^a^
**Behavioral intentions (never users)**		
Try EVP soon	158	0.52 (0.19–1.40)
Try EVP in next year	158	0.67 (0.27–1.63)
Try cigarette soon	185	0.55 (0.23–1.36)
Try cigarette in next year	185	0.53 (0.22–1.31)
**Behavior**		
Trial since baseline (never users)		
EVPs	190	1.27 (0.58–2.76)
Cigarettes	327	1.19 (0.59–2.40)
Past 30-day use ^b^		
EVPs	511	0.91 (0.49–1.70)
Cigarettes	510	1.80 (0.83–3.89)
Cigar/cigarillo/little cigar		
Quit or cut down in past month		
EVPs	91	0.95 (0.41–2.23)
Cigarettes	67	0.74 (0.28–1.96)

^a^ All models control for baseline measures of the outcome and baseline exposure to the Real Cost campaign. ^b^ Hookah and smokeless tobacco use not presented due to low baseline prevalence (<3%).

**Table 5 ijerph-19-14182-t005:** Association between mean perceived message effectiveness (PME) and vaping-related beliefs, harm perceptions, norms, behavioral intentions, and behavior immediately post-exposure and/or at 1-month follow-up in the intervention condition, PACE Vape Messaging Study, 2020.

	Post-Exposure		1-Month Follow-Up	
	N	OR (95% CI)	N	OR (95% CI)
**Vaping-related beliefs**				
Nicotine is main substance in EVPs that makes people want to vape (true vs. false/don’t know)	294	0.74 (0.52–1.06)	267	1.04 (0.71–1.52)
One 5% vape pod contains as much nicotine as pack of cigarettes (true vs. false/don’t know)	294	1.31 (0.99–1.72)	268	0.96 (0.69–1.33)
Addiction to nicotine is something I am concerned about (true vs. false/don’t know)	294	1.44 (1.10–1.90)	268	0.90 (0.68–1.20)
**Vaping-related harm perceptions**				
Harm from EVPs ^a^ (range 1 (no harm)–4 (a lot of harm))	294	0.056 (−0.025–0.14)	268	0.14 (0.051–0.22)
Harm of EVPs vs. smoking cigarettes ^b^	294		268	
Less harmful		0.64 (0.48–0.85)		1.10 (0.77–1.57)
No different		Ref.		Ref.
More harmful		0.78 (0.57–1.06)		1.14 (0.80–1.64)
Harm of vaping nicotine vs. marijuana/THC ^b^	294		268	
Less harmful		0.63 (0.45–0.87)		0.97 (0.67–1.40)
No different		Ref.		Ref.
More harmful		0.87 (0.66–1.15)		1.21 (0.90–1.63)
**Vaping-related norms (1 (very positive) to 5 (very negative)**				
Most people’s opinion of using EVPs like JUUL ^a^	293	−0.024 (−0.15–0.10)	268	−0.076 (−0.21–0.060)
Opinion of people who are important to you of using EVPs like JUUL ^a^	294	0.061 (−0.070–0.19)	266	−0.031 (−0.17–0.11)
**Potential spillover effects**				
Nicotine is cause of cancer (false vs. true/don’t know)	294	0.75 (0.56–1.01)	268	0.62 (0.45–0.86)
Flavored tobacco/EVPs easier to use vs. unflavored (vs. about the same/harder)	293	0.85 (0.65–1.12)	268	0.77 (0.57–1.04)
Flavored tobacco/EVPs is less harmful vs. unflavored (vs. about the same/more harmful)	294	1.04 (0.57–1.90)	268	0.78 (0.34–1.81)
A tobacco product that says it has no additives is less harmful than a regular tobacco product (false vs. true/don’t know)	294	1.10 (0.86–1.39)	268	1.02 (0.78–1.33)
A tobacco product that says it is organic tobacco is less harmful than a regular tobacco product (false vs. true/don’t know)	294	0.86 (0.67–1.11)	268	0.83 (0.62–1.11)
**Behavioral intentions (never users)**				
Try EVP soon	-	-	83	0.82 (0.39–1.73)
Try EVP in next year	-	-	83	0.75 (0.38–1.51)
Try cigarette soon	-	-	213	0.52 (0.35–0.79)
Try cigarette in next year	-	-	213	0.47 ** (0.31–0.72)
**Behavior**				
Trial since baseline (never users)				
EVPs	-	-	101	0.93 (0.53–1.60)
Cigarettes	-	-	165	1.11 (0.62–2.01)
Past 30-day use ^c^				
EVPs	-	-	268	1.07 (0.66–1.75)
Cigarettes	-	-	267	0.93 (0.53–1.66)
Cigar/cigarillo/little cigar			267	0.28 (0.12–0.67)
Quit or cut down in past month				
EVPs	-	-	35	1.27 (0.57–2.85)
Cigarettes	-	-	21	7.55 (0.83–69.0)

** Bonferroni-corrected *p* < 0.001. ^a^ Linear regression models; coefficients are betas (b). ^b^ Multinomial logistic regression models; coefficients are relative risk ratios (RRR). ^c^ Hookah and smokeless tobacco use not presented due to low baseline prevalence (<3%).

## Data Availability

The data presented in this study are available on request from the corresponding author. The data are not publicly available due to the sensitivity of data collected on tobacco use.

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
