# Peer review of "Effects of Vaping Prevention Messages on Electronic Vapor Product Beliefs, Perceived Harms, and Behavioral Intentions among Young Adults: A Randomized Controlled Trial"

_ijerph, 2022, doi:10.3390/ijerph192114182_

Round 1

Reviewer 1 Report

This study contributes to the literature on evaluation of health campaigns.

Reviewer 2 Report

ms #: ijerph-1911785 – Effects of Vaping Prevention Messages on Electronic Vapor Product Beliefs, Perceived Harms, and Behavioral Intentions Among Young Adults: A Randomized Controlled Trial; Villanti et al. 

 This study used an online messaging intervention in N=569 young adults residing in Vermont to examine the effect of vaping- versus sun safety-related messages on message perceptions, perceived message effectiveness (PME), electronic vaping product (EVP)-related beliefs, and EVP-related harm perceptions. Measures were collected again at 1-month follow-up in addition to any change in tobacco and EVP-related use and use intentions. They report finding that the vaping messaging intervention had some positive impacts, including higher PME, decreased curiosity to vape, and increased desire to quit or cut down on vaping. However, the intervention did not affect vaping-related beliefs, harm perceptions, norms, behavioral intentions, or behaviors.  Exploratory analyses in the intervention condition found that greater PME was associated with higher perceived harm of EVPs and a false belief about nicotine causing cancer.

 While this study addresses important issues related to public health prevention interventions to reduce vaping and other forms of tobacco/nicotine use in youth and young adults, I have some comments and questions that might help to clarify the manuscript and its conclusions.

1.     Many measures were utilized in this study. Could the authors state the primary, secondary, and exploratory hypotheses and outcomes earlier in the manuscript? It seems like they are being given equal weight across the manuscript. How variables are being conceptualized and prioritized is hard to track across the manuscript at times.

2.     How did the authors control for multiple statistical tests?

3.     Is it possible that there were ceiling effects in this study?

4.     Did the authors have enough statistical power to detect differences between groups?

5.     If the outcome in Table 5 is continuous, why are the authors using logistic regression analyses?

6.     Women represent 70% of the study sample. Why do the authors think this is the case and how might that affect generalizability of results?  
